# Measuring Diversity: Axioms and Challenges

**Mikhail Mironov** [1]  **Liudmila Prokhorenkova** [1]

## Abstract

This paper addresses the problem of quantifying diversity for a set of objects. First, we conduct a systematic review of existing diversity measures and explore their undesirable behavior in certain cases. Based on this review, we formulate three desirable properties (axioms) of a reliable diversity measure: monotonicity, uniqueness, and continuity. We show that none of the existing measures has all three properties and thus these measures are not suitable for quantifying diversity. Then, we construct two examples of measures that have all the desirable properties, thus proving that the list of axioms is not self-contradictory. Unfortunately, the constructed examples are too computationally expensive (NP-hard) for practical use. Thus, we pose an open problem of constructing a diversity measure that has all the listed properties and can be computed in practice or proving that all such measures are NP-hard to compute.

## 1. Introduction

Diversity of a collection of objects is a concept that is widely used in practice: image generation models are required to generate a diverse sample of images for a given prompt, recommender systems are required to output a diverse set of suggestions for a query, molecule generation models often aim at generating a collection of structurally diverse molecules with a given property. Diversity can also play an important role in assessing how representative a given dataset is, e.g., in molecule generation (Xie et al., 2023) or neural algorithmic reasoning (Veličković & Blundell, 2021; Mahdavi et al., 2023); and Zhao et al. (2024) argue that it is critical to provide a clear definition of diversity when analyzing datasets. A well-defined diversity measure also ensures informative sample selection in active learning (Ren et al., 2021) or may guide the generation of meaningful

[1]Yandex Research. Correspondence to: Mikhail Mironov <mironov.m.k@gmail.com>, Liudmila Prokhorenkova <ostroumova-la@yandex-team.ru>.

*Proceedings of the 42nd International Conference on Machine Learning*, Vancouver, Canada. PMLR 267, 2025. Copyright 2025 by the author(s).

synthetic data in augmentation (Mumuni & Mumuni, 2022). Thus, being able to quantify diversity is important.

Traditional methods of assessing diversity may differ across domains and tasks. In the image generation domain, diversity ensures that at least some of the generated images can fit a user's preference. The average of pairwise distances between the output images is commonly used as a measure of diversity. For instance, Ruiz et al. (2023) compute diversity as the average LPIPS similarity between the output objects, while Saharia et al. (2022) compute the average pairwise SSIM between the first output sample and the remaining samples. Similarly, in recommender systems, diversity ensures that at least some of the model outputs can fit a user's preference. The average pairwise distance between the outputs is a popular diversity measure in this domain (Alhijawi et al., 2022). Another way of assessing diversity is via the determinantal point process (DPP) approach that defines diversity as the determinant of the similarity matrix (Wilhelm et al., 2018). In the molecule generation domain, the typical task is to generate a diverse collection of molecules with some predefined properties. The underlying goal is to explore the whole space of such possible molecules and pick the best candidates, so diversity of the output collection ensures that generated molecules are not clustered in one area, while other areas are unexplored. A common diversity measure here is also the average pairwise distance between the outputs (Du et al., 2022), although sometimes the percentage of unique generated molecules is reported (Hoogeboom et al., 2022). Finally, in a recent paper on generating structurally diverse graphs (Velikonivtsev et al., 2024), a new measure called *energy* is proposed as a better and more reliable alternative to the average pairwise distance.

Note that in all the examples above, diversity can also be thought of as *coverage*: the goal is to cover different areas of the space of potentially valid outputs. Thus, in this paper, we use the terms *diversity* and *coverage* interchangeably. In the literature, there have been a few attempts to analyze, compare, or suggest better measures of diversity (Xie et al., 2023; Friedman & Dieng, 2023; Velikonivtsev et al., 2024). However, as we show in this paper, the problem is still underexplored.

We limit the scope of our research to the following setup: we are given a collection of abstract objects and their pairwise

distances (or pairwise similarities). We define a diversity measure as a function that takes this collection as an input and returns some value as an output.

First, we examine the existing diversity measures by providing examples of their undesirable behavior. Namely, we show that existing measures may either lead to unexpected results when comparing diversity of two datasets (i.e., assigning a higher score to a clearly less diverse dataset) or lead to degenerate solutions when being optimized. Motivated by these observations and previous studies on diversity, we formulate three properties (axioms) that a good diversity measure should have. *Monotonicity* requires that increasing pairwise distances between the objects increases diversity value. *Uniqueness* requires that having a duplicate in the collection is worse for diversity than having any non-duplicate object instead. The last property is *continuity*, which requires diversity to be a continuous function of pairwise distances. We check which of the existing measures possess which properties, and find that none has all three. Then, we prove that the list of axioms is not self-contradictory by constructing two examples of measures that satisfy all of them. Unfortunately, the proposed measures are too computationally expensive (NP-hard) to be used in practice. Finally, we discuss why finding a diversity measure that has all three desirable properties and is computationally manageable is a non-trivial task. We leave the question of whether there exists a computationally feasible measure satisfying all the required axioms for future studies.

## 2. Measuring Diversity

In this section, we describe existing diversity measures. We assume that we are given a collection of $n$ (possibly duplicated) objects $X = (x_1, \ldots, x_n)$ and pairwise distances (dissimilarities) between them such that $d_{ij} \geq 0$ and $d_{ij} = 0$ iff $x_i$ and $x_j$ coincide. To maintain generality, we do not require the triangle inequality to be satisfied by $d_{ij}$.

Table 1 lists existing diversity measures that we cover in our study. As discussed above, arguably the most straightforward and widely-used way to quantify diversity is via the *average pairwise distance* between the elements. Other simple alternatives are the *minimum* and *maximum* pairwise distances (often referred to as Bottleneck and Diameter, respectively). Xie et al. (2023) argue that none of the simple measures are suitable for diversity quantification and propose #Circles($t$) which is defined as the maximum number of non-intersecting circles of radius $t/2$ (for some $t > 0$) with centers at some elements of $X$. A measure called Energy($\gamma$) is proposed by Velikonivtsev et al. (2024) as a better alternative to the above measures. For $\gamma = 1$, this measure equals the energy of a system of equally charged particles. Hu et al. (2024) propose measuring diversity as the length of the shortest Hamiltonian circuit; we further

*Table 1.* Known diversity measures

| Measure | Formula |
| --- | --- |
| Average | $\frac{2}{n(n-1)} \sum_{i<j} d_{ij}$ |
| SumAverage | $\frac{1}{n} \sum_{i<j} d_{ij}$ |
| Diameter | $\max_{i<j} d_{ij}$ |
| SumDiameter | $\sum_i \max_{j\neq i} d_{ij}$ |
| Bottleneck | $\min_{i<j} d_{ij}$ |
| SumBottleneck | $\sum_i \min_{j\neq i} d_{ij}$ |
| Energy($\gamma$), $\gamma > 0$ | $-\frac{2}{n(n-1)} \sum_{i<j} \frac{1}{d_{ij}^\gamma}$ |
| #Circles($t$), $t \geq 0$ | $\max_{C\subseteq[n]} |C|$ s.t. $d_{ij}>t \, \forall \, i\neq j \in C$ |
| Unique | $\max_{C\subseteq[n]} \frac{|C|}{n}$ s.t. $d_{ij}>0 \, \forall \, i\neq j \in C$ |
| HamDiv | length of the shortest Hamiltonian circuit |
| Vendi Score | $\exp\left(-\sum_i \lambda_i \log(\lambda_i)\right)$ |
| DPP | $\det(S)$ |
| RKE | $-\log\left(\frac{1}{n^2} \sum_{i,j} s_{ij}^2\right)$ |
| Species($q$), $1\neq q\geq 0$ | $\left(\sum_i \left(\sum_j s_{ij}\right)^{q-1}\right)^{\frac{1}{1-q}}$ |

refer to this measure as HamDiv.

The remaining four measures are defined in terms of pairwise similarities $s_{ij}$ instead of pairwise distances. All these measures require $s_{ij}$ to be a positive semi-definite similarity function and usually require $s_{ii} = 1$. Vendi Score is proposed by Friedman & Dieng (2023) and is calculated via the formula specified in Table 1, where $\lambda_1, \ldots, \lambda_n$ are the eigenvalues of the scaled similarity matrix $S/n$ and $S$ is the $n \times n$ matrix with entries $s_{ij}$. The simplest DPP-based measure is computed as the determinant of the similarity matrix $S$.[1] The Rényi Kernel Entropy Mode Count (RKE) is proposed by Jalali et al. (2023) and is defined as the negative logarithm of the average squared similarity. Finally, *diversity of order* $q$ is proposed by Leinster & Cobbold (2012) to measure the diversity of a population consisting of several species. In our work, we refer to this measure as Species($q$). Here, the parameter $q$ is any nonnegative number not equal to 1. When applied to our setup (all elements having equal weights), the measure Species($q$) can be written as specified in Table 1 (up to a constant multiplier).

Some previous works on measuring diversity analyze and compare measures based on properties they do or do not satisfy. We review these works in Section 4.4.

---

[1] In practice, more complex DPP-based diversity measures can be used (Wilhelm et al., 2018). For instance, when such measures are applied to recommender systems, the relevance scores of objects w.r.t user queries are usually mixed into the similarity matrix, which we do not do here since we only consider diversity.

Finally, there is a rich literature on submodular functions that can also be used to quantify diversity (Bilmes, 2022). However, submodular functions are not defined in terms of pairwise distances and thus are out of scope of our study. Also, some known submodular functions assume that we are provided with a set (space) of all possible objects and can sum, integrate, or iterate over all of them. This assumption is quite restrictive and our framework does not rely on it.

## 3. Drawbacks of Popular Diversity Measures

In this section, we discuss why none of the measures defined above can be reliably used to quantify diversity. For this, we show intuitive examples of an undesirable behavior for each measure. These examples serve as the main motivation for our research and for the axioms we choose.

We start by discussing two usage scenarios of diversity measures. First, a diversity measure can be applied to a given dataset to quantify its diversity. Thus, it should be able to identify which dataset is more diverse. For instance, when choosing between two recommendation algorithms, one can be interested in comparing the diversity of the retrieved sets of items. Second, diversity can be used as a goal of an optimization process. For instance, Velikonivtsev et al. (2024) generate sets of graphs that are maximally diverse and for this purpose, the authors iteratively modify the set of graphs by accepting modifications that improve a given diversity measure. Thus, a good diversity measure should lead to diverse configurations of elements when being optimized.

Below we examine the diversity measures listed in Table 1 from these two perspectives: *comparison* and *optimization*. We say that a measure exhibits undesirable behavior w.r.t. *comparison* if there exists a pair of datasets, such that the first one is more diverse according to our intuitive perception of diversity, yet the diversity measure assigns the higher value to the second one. We say that a measure exhibits undesirable behavior w.r.t. *optimization* if the dataset with the maximum diversity according to this measure is not maximally diverse according to our intuitive perception of diversity. Note that if a measure exhibits undesirable behavior w.r.t. optimization, it also exhibits undesirable behavior w.r.t. comparison. Indeed, if a measure assigns the highest value to some intuitively non-diverse set, this means that it assigns a lower value to some set that is more intuitively diverse, thus exhibiting undesirable behavior w.r.t. comparison. The opposite is not necessarily true: some measures can be suitable for optimization while being unable to reliably compare two non-optimal configurations.

Note that we limit our research to the simple case when the number of elements $n$ is fixed; thus, in the examples below all the configurations are of the same size.

**Average and SumAverage**  Since Average and SumAverage differ only by a constant factor, we consider them together. Consider two configurations of 16 points in the unit square with Euclidean distance, as illustrated in the figure below (in the configuration on the left, each of the square's corners contains 4 coinciding points). For the left configuration, Average equals 0.91, which is the maximum value among all possible configurations. For the right configuration, Average equals 0.71. Since the right configuration is intuitively more diverse, this example shows undesirable behavior of Average w.r.t. both comparison and optimization. Informally, maximizing Average pushes all points to the boundary of the space, leaving central areas empty.

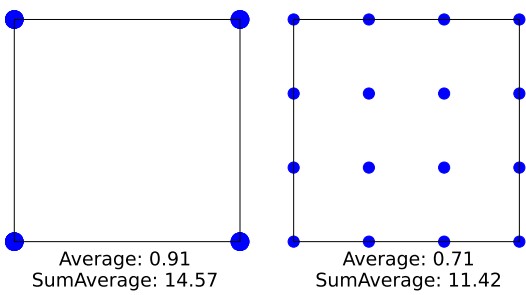

Average: 0.91
SumAverage: 14.57          Average: 0.71
                           SumAverage: 11.42

**Diameter and SumDiameter**  Consider two configurations of 16 points in the unit square as shown below (in the left configuration, two of the square's corners contain 8 coinciding points each). Diameter for both configurations equals 1.41, which is the maximum value among all possible configurations. Since the right configuration is intuitively more diverse, this example shows undesirable behavior of Diameter w.r.t. both comparison and optimization. Note that once a configuration contains two points at the maximum distance from each other (in our case 1.41), the positions of all other points do not influence Diameter. While SumDiameter is expected to be a better diversity measure (it takes more distances into account), the same example demonstrates its undesirable behavior w.r.t. comparison and optimization since the left configuration has the maximum possible SumDiameter value. Indeed, if there are points $x_1$ and $x_2$ with the maximum distance between them, we can make all other points coincide with $x_1$ or $x_2$, thus maximizing SumDiameter.

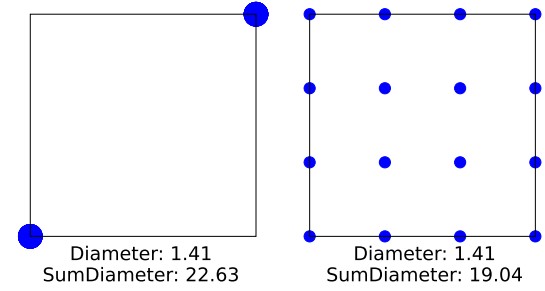

Diameter: 1.41
SumDiameter: 22.63          Diameter: 1.41
                           SumDiameter: 19.04

**Bottleneck**    Bottleneck assigns any configuration without duplicates a higher diversity value than any configuration with duplicates. Consider two configurations of 16 points in the unit square (in the right configuration of the figure below, the bottom-left corner contains 2 coinciding points). For the left configuration, Bottleneck equals $0.11$, and for the right configuration, Bottleneck equals $0$. Since the right configuration is intuitively more diverse, we see undesirable behavior of Bottleneck w.r.t. comparison.

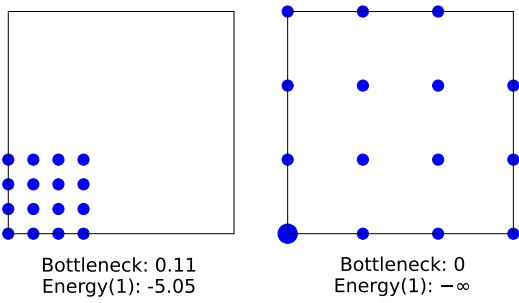

Bottleneck: 0.11
Energy(1): -5.05

Bottleneck: 0
Energy(1): $-\infty$

**SumBottleneck**    To a lesser extent, SumBottleneck has similar drawbacks to Bottleneck. Consider two configurations of 16 points in the unit square (in the left configuration, 15 points coincide in a corner of the square, and in the right configuration, each point has one duplicate). For the left configuration, SumBottleneck equals $0.1$, and for the right configuration, SumBottleneck equals $0$. Since the right configuration is intuitively more diverse, we see undesirable behavior of SumBottleneck w.r.t. comparison.

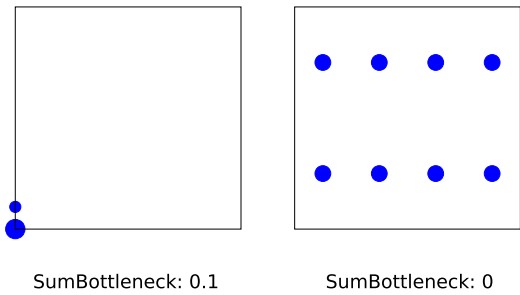

SumBottleneck: 0.1

SumBottleneck: 0

**Energy**$(\gamma)$    The drawback of this measure is that in the presence of a duplicate, it takes the value $-\infty$ and is insensitive to all other pairwise distances. The same example as for Bottleneck demonstrates undesirable behavior of Energy w.r.t. comparison.

Note that the examples for Bottleneck, SumBottleneck, and Energy demonstrate their undesirable behavior only w.r.t. comparison. Intuitively, all these measures behave well w.r.t. optimization since maximizing them enforces a more uniform distribution by pushing away the closest elements (the examples for Energy optimization can be found in Velikonivtsev et al. (2024)).

**#Circles**$(t)$    To use this measure for a reasonable comparison of two collections, one needs to determine an appropriate value of $t$. Indeed, if $t$ is too high, both collections will have diversity equal to $1$, and if $t$ is too low, both collections will have diversity equal to their number of unique elements. This complicates the use of this measure for both comparison and optimization. Also, this measure is discrete and thus difficult to optimize. Finally, computing the value of this measure is NP-hard, which makes it impractical.

**Unique**    Since this measure does not take into account pairwise distances between objects, it is essentially unsuitable for comparison or optimization. Indeed, all collections with pairwise distinct objects have the same diversity value $1$.

**HamDiv**    Similar to #Circles$(t)$, the value of HamDiv is NP-hard to compute, which makes it impractical. Additionally, this measure is insensitive to increases in any pairwise distances that are not part of the shortest Hamiltonian circuit, making it unsuitable for comparison. To illustrate its undesirable behavior w.r.t. optimization, consider the following example. Take two configurations of $4$ points on the line segment $[0, 1]$: the first is $0, 0, 1, 1$, and the second is $0, \frac{1}{3}, \frac{2}{3}, 1$. Both configurations have HamDiv equal to $2$, which is the maximum possible value. However, the first collection is intuitively less diverse than the second.

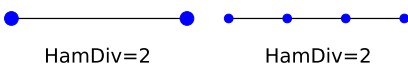

HamDiv=2          HamDiv=2

**Vendi Score**    Consider points on a circle with cosine similarity. Suppose the points $x_1, x_2, x_3$ are arranged in this order on the circle; the distance from $x_1$ to $x_2$ is $0.6$ radians, and the distance from $x_2$ to $x_3$ is $1.4$ radians. Now, we move $x_3$ by $0.1$ radians further away from $x_1$ and $x_2$. Intuitively, we expect that decreasing the similarity between $x_3$ and the other elements should increase diversity. However, the Vendi Score decreases from $1.941$ to $1.916$, which is an example of undesirable behavior w.r.t. comparison.

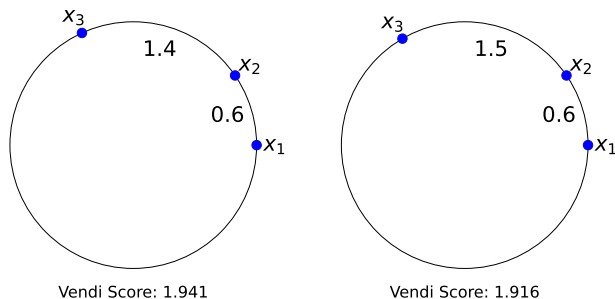

Vendi Score: 1.941          Vendi Score: 1.916

**DPP**   Consider two positive semidefinite symmetric matrices:

$$S = \begin{pmatrix} 1 & 0.2 & 0.6 \\ 0.2 & 1 & 0.7 \\ 0.6 & 0.7 & 1 \end{pmatrix}, \quad \hat{S} = \begin{pmatrix} 1 & 0.3 & 0.6 \\ 0.3 & 1 & 0.7 \\ 0.6 & 0.7 & 1 \end{pmatrix}.$$

The matrices $S$ and $\hat{S}$ differ by an increase in $s_{12}$ (and symmetrically $s_{21}$) from 0.2 to 0.3. Intuitively, we expect that increasing similarity between any two elements must decrease diversity. However, $\det(S) = 0.278$ and $\det(\hat{S}) = 0.312 > 0.278$, which is an example of undesirable behavior of the DPP-based measure w.r.t. comparison.

**RKE and Species**($q$)   Consider points on a circle with cosine similarity (see the figure below for an illustration). Suppose the points $x_1, x_2, x_3$ are arranged in this order on the circle; the distance from $x_1$ to $x_2$ is 1.1 radians, the distance from $x_2$ to $x_3$ is 0.4 radians. Now, we make $x_2$ a duplicate of $x_3$. Intuitively, we expect that such a change should decrease diversity. However, RKE increases from 0.564 to 0.584, which is an example of undesirable behavior w.r.t. comparison. The same example illustrates undesirable behavior w.r.t. comparison for Species($q$) for various $q$ (see Appendix B).

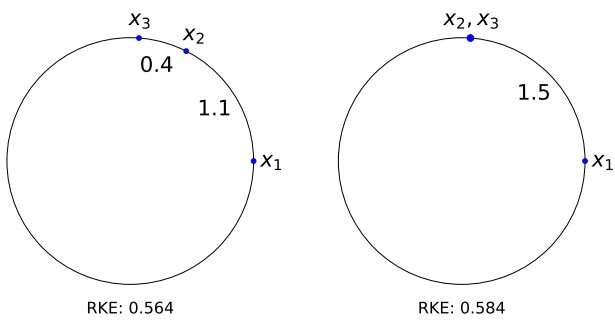

RKE: 0.564          RKE: 0.584

# 4. Axiomatic Approach to Diversity Measures

Motivated by our analysis in Section 3, we formulate a list of properties (axioms) that a reliable diversity measure is expected to satisfy. First, we formally define diversity measures, then formulate their desirable properties and discuss which existing measures have which properties, and finally review desirable properties suggested in previous studies and discuss how they relate to our setup.

## 4.1. Formal Definition of Diversity Measure

Assume that we are given a collection of $n$ (possibly duplicated) objects $X = (x_1, \ldots, x_n)$ and pairwise distances between them $d_{ij}$ satisfying the following conditions:

1. $\forall i, j : d_{ij} \geq 0$ and $\forall i : d_{ii} = 0$;

2. if $d_{ij} = 0$, then $\forall k : d_{ik} = d_{jk}$;

3. $\forall i, j : d_{ij} = d_{ji}$.

In terms of objects, the first property requires that the distance between any two objects is nonnegative, and distance from an object to itself is 0. The second property requires that if two objects coincide, then they must have equal distances to any other object. The third property is symmetry of distance. Note that for generality, we do not require the triangle inequality to be satisfied by $d_{ij}$.

A diversity measure is a function that takes as input any such set of $n$ objects and their pairwise distances and outputs a real number. We assume that diversity depends only on distances $d_{ij}$ and does not depend on the nature of the objects $x_i$ itself. Thus, the input of our function can be fully described as an $n \times n$ matrix $D$ with entries $d_{ij}$. Denote by $D_n$ a subset of all $n \times n$ matrices satisfying the three properties described above. Then, the diversity function is a function from $D_n$ to $\mathbb{R}$. Since diversity is usually measured for a *multiset* of objects, we also require *permutation invariance*: if we permute (or rename) the objects in $X$ (with correspondingly permuting the rows and columns of $D$), the value of diversity should not change. Thus, we get the following definition.

**Definition 4.1.** A *diversity function* is a permutation-invariant function from $D_n$ to $\mathbb{R}$.

Note that we assume the number of elements $n$ to be fixed. Thus, we do not aim to determine how diversity should behave when the size of the dataset changes. Our paper shows that even for this (simpler) case it is non-trivial to construct a suitable diversity measure.

## 4.2. Axioms for Diversity

In this section, we formulate three axioms that we require for a reliable diversity measure.

**Axiom 1** (Monotonicity). *A diversity function must be strictly increasing w.r.t. all its arguments.*

In other words, if we increase one or several pairwise distances while keeping all other distances fixed, the value of diversity must increase. This axiom is natural since it represents the meaning of diversity: the more objects $x_1, \ldots, x_n$ differ from each other, the greater diversity we expect. This property is analogous to monotonicity in Velikonivtsev et al. (2024), but has one important difference: we do not require the objects in $X$ to be pairwise distinct for monotonicity to hold. This difference is critical for being able to compare datasets: we want to be able to tell which configuration is more diverse even if the datasets have duplicates. Otherwise, we may get a measure with undesirable behavior, as shown by the example for Bottleneck and Energy in Section 3.

**Axiom 2** (Uniqueness). *Suppose we are given two collections of objects (and their pairwise distances) which differ only by one element: $x_1, \ldots, x_{n-1}, x_n$ and $x_1, \ldots, x_{n-1}, x_n'$. Suppose $x_n'$ coincides with at least one of $x_1, \ldots, x_{n-1}$, while $x_n$ does not coincide with any of $x_1, \ldots, x_{n-1}$. Then, the diversity of the first collection must be higher than that of the second collection.*[2]

This property reflects our intuition that having a duplicate ($x_n'$) in the multiset is worse for diversity than having a unique element ($x_n$) instead. Informally, we can say that having $x_n'$ does not help the multiset cover any new part of the space since a copy of $x_n'$ is already present, while having $x_n$ covers some new area. Uniqueness allows one to avoid an undesirable behavior when the collection with duplicates has higher diversity than an intuitively more diverse collection without duplicates or even when the maximum diversity is achieved by a degenerate configuration (which happens to Average and Diameter, as shown in Section 3). Note that, unlike the analogous property in Velikonivtsev et al. (2024), our variant of uniqueness does not require all objects in $X$ to be distinct. Similar to considerations for monotonicity, this modification is important for being able to compare datasets even when they have duplicated elements.

**Axiom 3** (Continuity). *A diversity function must be continuous.*

This property was not present in previous works, but it is natural to require, and we find it critical for a reliable diversity measure. Indeed, in Appendix A.2 we show that there are examples of discontinuous functions that satisfy monotonicity and uniqueness while still exhibiting undesirable behavior. Thus, having only monotonicity and uniqueness is insufficient.

### 4.3. Properties of Existing Measures

Table 2 shows which axioms are satisfied by the existing measures listed in Table 1 (the proofs can be found in Appendix B). It can be seen that none of these measures has all three desirable properties.[3] This leads us to the main question of the paper: does there exist a diversity measure with all three desirable properties? In the next section, we construct two examples of such measures, thus giving a positive answer to this question. We include these measures as well as the computational complexities of all the measures in Table 2.

---

[2]For simplicity, we formulate this property in terms of objects, but it can be straightforwardly reformulated in terms of pairwise distances, see Appendix A.1.

[3]Note that Energy was reported in Velikonivtsev et al. (2024) as having monotonicity and uniqueness, but it does not in our case since we have stronger versions of these properties that require them to hold even in the presence of duplicated elements.

### 4.4. Desirable Properties in Previous Works

Several papers analyze and compare diversity measures in terms of properties they do or do not satisfy. For instance, Xie et al. (2023) formulate three axioms. The first one requires that diversity of a union of two sets must be higher than the diversity of each of them. The second requires that diversity of a union of two sets should be at most the sum of their diversities. Note that both of these axioms constrain the behavior of diversity when the number of objects changes and thus do not apply in our setting with a fixed number of objects. The last axiom requires that if we have only two objects in $X$, then diversity must be strictly monotone w.r.t. the pairwise distance between these objects. Note that our monotonicity requirement generalizes this axiom.

Friedman & Dieng (2023) propose Vendi Score and list four of its properties. One of the properties is called *symmetry* and it is equivalent to our *permutation invariance* that we require for all diversity measures. Another property requires that a diversity measure is maximized when all pairwise similarities are 0 and minimized when all pairwise similarities are 1. This property is generalized by our monotonicity axiom. The remaining two properties consider weighted elements or samples of different sizes and thus do not apply to our setup.

Velikonivtsev et al. (2024) address the problem of generating structurally diverse graphs and discuss what measures of diversity are suitable for optimization. The authors formulate two properties: *monotonicity* and *uniqueness* that are slightly weaker variants of the corresponding properties in our work (as discussed above in Section 4.2).

Leinster & Cobbold (2012) list several groups of properties of Species($q$). *Partitioning properties* do not apply to our case since we consider the diversity only for a fixed number of objects. From the *Elementary properties* group, Symmetry corresponds to our requirement for diversity functions to be permutation invariant, and the properties Absent species and Identical species do not apply to our case (since we consider $n$ objects with equal weight and not $n$ probabilities summing to 1). From the group of properties named *Effect of species similarity on diversity*, the only property applicable in our case is Monotonicity, which is equivalent to our monotonicity axiom.

To sum up, among the properties from previous works, the ones applicable in our setting are monotonicity (in stronger form from Velikonivtsev et al. (2024) and Leinster & Cobbold (2012) or weaker forms from Xie et al. (2023) and Friedman & Dieng (2023)) and uniqueness, given that permutation invariance is already incorporated in our definition of a diversity function.

*Table 2.* Properties of diversity measures

| Measure | Monotonicity | Uniqueness | Continuity | Complexity |
|---|:---:|:---:|:---:|:---:|
| Average | ✓ | ✗ | ✓ | $O(n^2)$ |
| SumAverage | ✓ | ✗ | ✓ | $O(n^2)$ |
| Diameter | ✗ | ✗ | ✓ | $O(n^2)$ |
| SumDiameter | ✗ | ✗ | ✓ | $O(n^2)$ |
| Bottleneck | ✗ | ✗ | ✓ | $O(n^2)$ |
| SumBottleneck | ✗ | ✗ | ✓ | $O(n^2)$ |
| Energy$(\gamma)$, $\gamma > 0$ | ✗ | ✗ | ✓ | $O(n^2)$ |
| #Circles$(t)$, $t \geq 0$ | ✗ | ✗ | ✗ | NP-hard |
| Unique | ✗ | ✓ | ✗ | $O(n)$ |
| HamDiv | ✗ | ✗ | ✓ | NP-hard |
| Vendi Score | ✗ | ✗ | ✓ | $O(n^3)$ |
| DPP | ✗ | ✗ | ✓ | $O(n^3)$ |
| RKE | ✓ | ✗ | ✓ | $O(n^2)$ |
| Species$(q)$ | ✓ | ✗ | ✓ | $O(n^2)$ |
| MultiDimVolume | ✓ | ✓ | ✓ | NP-hard |
| IntegralMaxClique | ✓ | ✓ | ✓ | NP-hard |

## 5. Diversity Measures with All Desirable Properties

In this section, we construct two different examples of permutation-invariant measures that have all three desirable properties.

**MultiDimVolume**  For a given $k$, $2 \leq k \leq n$, and a given submultiset $K$ of size $k$ of the multiset $X = \{x_1, \ldots, x_n\}$, calculate the product of all pairwise distances between the elements of $K$. Note that this product equals zero if at least two elements of $K$ coincide. Then, for a given $k$, we take the maximum of such products over all submultisets of size $k$ of $X$ and denote this maximum as $m_k(X)$. We define the diversity of $X$ as $\sum_{k=2}^{n} m_k(X)$. Putting the above into one formula, we get:

$$\text{Diversity}(X) := \sum_{k=2}^{n} \max_{\substack{K \subseteq X \\ |K|=k}} \left( \prod_{\substack{x_i, x_j \in K \\ i < j}} d_{ij} \right). \quad (1)$$

The intuition behind this formula is that for a set $K$ of size $k$, the product of all pairwise distances between the elements of $K$ can be thought of as an analog of $k$-dimensional volume of $K$ (analogy comes from the fact that if two elements of $K$ coincide, then the volume equals zero). Thus, $m_k(X)$ is the maximum 'volume' of a $k$-dimensional subset of $X$.

In Appendix C, we prove that MultiDimVolume satisfies all the axioms. Unfortunately, computing $\text{Diversity}(X)$ in Equation (1) is NP-hard since calculating MultiDimVolume allows one to solve the problem of finding the size of the maximum clique in a graph, and this problem is known to be NP-hard. We refer to Appendix C for the formal proof.

Let us also note that there are multiple ways to define diversity based on the values $m_k(X)$. Indeed, we can consider $\sum_{k=2}^{n} f(m_k(X))$, where $f$ is an arbitrary continuous monotone function. In particular, one may consider $\text{Diversity}(X) = \sum_{k=2}^{n} m_k(X)^{\frac{2}{k(k-1)}}$. This modification is natural since each summand is a product of $k(k-1)/2$ terms. It follows from the proof in Appendix C that all such modifications satisfy all the desirable properties.

**IntegralMaxClique**  For a given threshold $t \geq 0$, we construct the following graph. The nodes are $x_1, \ldots, x_n$. Two nodes $x_i$ and $x_j$ are connected by an edge iff $d_{ij} \geq t$, and we assign $d_{ij}$ as the weight of this edge. We find a clique (complete subgraph) in this graph with the maximum number of nodes. If there are several such cliques, we pick the one with the maximum total weight of edges. For the chosen clique, we calculate the total weight of its edges and denote it by $w_t(X)$. Then, we define diversity as

$$\text{Diversity}(X) := \int_0^{+\infty} w_t(X)\, dt. \quad (2)$$

This integral is finite since $w_t(X)$ is bounded by $\sum_{i<j} d_{ij}$, and if $t > \max_{i<j} d_{ij}$, then the constructed graph has no edges and $w_t(X) = 0$.

The intuition behind this formula is that $w_t(X)$ can be interpreted as the maximum diversity of a subset of $X$ with the

restriction that its elements should be at distance $t$ or more from each other.

In Appendix D, we prove that IntegralMaxClique satisfies all the axioms. Unfortunately, computing $\text{Diversity}(X)$ in Equation (2) is NP-hard since, similarly to MultiDimVolume, calculating IntegralMaxClique allows one to solve the problem of finding the size of the maximum clique in a graph. We refer to Appendix D for the formal proof.

By constructing the two examples above, we prove that three desirable properties from our list do not contradict each other. Unfortunately, the constructed examples are too computationally complex for most practical applications.

## 6. Discussion

In the previous section, we prove that the three axioms listed in Section 4.2 do not contradict each other. However, we have not been able to construct a measure that satisfies these axioms and is computationally feasible for practical application. We pose this as an important open problem to be addressed in future studies.

Let us provide some intuition on why it is hard to combine monotonicity, uniqueness, and continuity in one function. We first formulate the following proposition that shows an additional restriction that these three axioms imply.

**Proposition 6.1.** *Suppose a diversity function has uniqueness and continuity. Let $x_1, \ldots, x_k$ be a set of $k$ pairwise different objects. Let $C$ be a multiset of $n - k$ objects, each of which coincides with one of $x_1, \ldots, x_k$. Then, diversity of the multiset $\{x_1, \ldots, x_k\} \cup C$ is the same for all such $C$.*

We prove this proposition in Appendix E.1. Informally, Proposition 6.1 states that the diversity of a set does not depend on which elements are duplicated. This agrees well with our intuition: duplicates do not provide any additional elements and thus are not supposed to affect diversity. On the other hand, constructing a measure that is continuous while 'ignoring' duplicates is tricky since the object's property of being a duplicate is discontinuous. Indeed, we can move a duplicate by any small $\epsilon > 0$ and it stops being a duplicate, so our measure should no longer 'ignore' it. In MultiDimVolume, we address this problem by incorporating products of pairwise distances within subgraphs: any duplicate zeros the corresponding products and thus the placement of a duplicate does not affect the result. In IntegralMaxClique, we use a threshold $t$ to filter out small edges, and thus duplicates do not affect the value for all $t > 0$.

The next proposition states that a diversity function satisfying all the axioms cannot be expressed in a certain form. This particular form is motivated by the approach in Velikonivtsev et al. (2024): the authors iteratively improve

diversity of a set by updating one element at a time. Thus, they decompose a considered diversity function into the fitness of one element and diversity of the rest of the elements. Such decomposition would allow one to make quick updates of diversity (in linear time) when only one element is updated. In the proposition below, we show that for a proper diversity measure such decomposition cannot exist if we assume *additive* aggregation.

**Proposition 6.2.** *Assume that a diversity function can be decomposed in the following way:*

$$\text{Diversity}(X) = F(d_{12}, d_{13}, \ldots, d_{1n}) + G(x_2, \ldots, x_n),$$

*that is, the first term depends only on distances from one object $x_1$ to all other objects, and the second term depends only on pairwise distances between the objects $x_2, \ldots, x_n$. Then, such a diversity function cannot simultaneously satisfy monotonicity, uniqueness, and continuity axioms.*

We prove this proposition in Appendix E.2. This is a negative result showing why it can be difficult to construct a proper diversity measure that is convenient for optimization. Note, however, that this proposition is only proven for the additive aggregations, thus other options are potentially possible.

**Diversity measures in practice** Let us note that even NP-hard diversity measures can still be used in practice if a set of items that need to be evaluated is sufficiently small. For instance, if a recommender service returns a set of $k = 100$ items and we want to measure diversity of this set, then an NP-hard measure having all the desirable properties can potentially be used. Examples of diversity measures constructed in Section 5 demonstrate that there are several different options that can be used (e.g., MultiDimVolume and IntegralMaxClique, along with their variations satisfying all the properties). We cannot rule out any of these measures based on their theoretical properties. Thus, a decision on which measure should be used may depend on a particular application. However, in this paper, we use the measures MultiDimVolume and IntegralMaxClique only to prove that our list of axioms is not self-contradictory. Evaluating these measures in practical applications goes beyond the scope of the current paper, which addresses the theoretical aspects of diversity measures applicable to a wide variety of scenarios.

## 7. Conclusion

In this paper, we reviewed existing diversity measures and demonstrated via intuitive examples that these measures cannot be reliably used for evaluating diversity. Based on these examples and previous research on diversity measures, we formulated three simple axioms (desirable properties) for a reliable diversity measure: monotonicity, uniqueness, and continuity. It turns out that none of the previously known

measures has all these properties. We constructed two diversity measures that have all the desirable properties, thus proving that the axioms do not contradict each other. Unfortunately, the constructed examples are too computationally complex for practical use.

We leave for future research an important open problem of constructing a diversity measure that has all three desirable properties and is computationally feasible or proving that such a measure cannot exist. While our study does not answer this question, we believe that it gives some important insights into measures of diversity that are frequently used in practice. Being aware of what shortcomings a particular measure has, one can use it more wisely. For instance, we cannot advise using Energy for comparing diversities of arbitrary datasets, while it can be safely used as a target for optimization.

## Impact Statement

This paper presents work whose goal is to advance the field of Machine Learning. There are many potential societal consequences of our work, none which we feel must be specifically highlighted here.

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

## A. Axioms for Diversity

### A.1. Formal Definitions

In this section, we provide the formal definitions of the introduced axioms. Recall that diversity is a permutation-invariant function on $D_n$.

**Axiom 1** (Monotonicity). *For any two matrices $A, B \in D_n$ such that $\forall i, j \ a_{ij} \geq b_{ij}$ and at least one of the inequalities is strict, we have $\text{Diversity}(A) > \text{Diversity}(B)$.*

**Axiom 2** (Uniqueness). *Suppose that for $A, B \in D_n$:*

- $a_{ij} = b_{ij}$ *for all $i > 2, j > 2$;*

- $b_{1j} = b_{2j}$ *for all $j$;*

- $a_{2j} = b_{2j}$ *for all $j > 2$;*

- $a_{1j} > 0$ *for all $j > 1$.*

*Then,* $\text{Diversity}(A) > \text{Diversity}(B)$.

**Axiom 3** (Continuity). *The space $D_n$ is endowed with the subspace topology induced by the standard topology on the set of all $n \times n$ matrices. A diversity function must be continuous w.r.t. this topology.*

### A.2. The Necessity of Continuity Axiom

Consider any measure $M$ that is monotone (e.g., Average). Let $M'$ be an order-preserving transformation of $M$ whose range is $[0, 1)$ (e.g., if $M$ takes only non-negative values, we can take $M'(X) := 1 - e^{-M(X)}$). Then, the measure $\text{Unique}(X) + M'(X)$ has both uniqueness and monotonicity. Essentially, $\text{Unique}(X) + M'(X)$ compares two configurations in the following way:

- Count the number of unique elements in both configurations;

- If the numbers of unique elements are different, then the configuration with a bigger number is more diverse;

- If the numbers of unique elements are the same, compare configurations based on the measure $M$ (or, equivalently, $M'$).

We argue that $\text{Unique}(X) + M'(X)$ is not a good diversity measure. To illustrate, consider any measure $M$ that has the monotonicity property (for instance Average), optimize it, and after that spread the elements a bit to make them unique. Then, we get a (nearly) optimal configuration for $\text{Unique}(X) + M'(X)$ which is very similar to the optimal configuration for $M(X)$. However, the optimal solution for $M(X)$ may be not diverse, as we show with the example for Average in Section 3.

The discreteness of $\text{Unique}(X)$ plays a crucial role in the construction above. A natural way to prevent the measures of the form $\text{Unique}(X) + M'(X)$ from being considered good diversity measures is to require that diversity measures must be continuous.

## B. Properties of Diversity Measures: Proofs

Let us prove the statements about which measures possess which properties, as indicated in Table 2. Note that for some of the measures, their monotonicity and uniqueness were analyzed by Velikonivtsev et al. (2024). However, since we modified these properties, we need to formally check the new ones.

**Average and SumAverage** Monotonicity and continuity are trivial, as is the $O(n^2)$ complexity. To prove that uniqueness does not hold, consider the example from Section 3: given 16 points in a square with Euclidean distance, the maximum diversity is achieved when every vertex contains 4 objects, and replacing any of these duplicates by any other object will decrease diversity.

**Diameter and SumDiameter**   Consider a collection of three objects with pairwise distances $2, 2, 1$. Increasing distance $1$ to $2$ does not change the diversity value, thus proving that monotonicity does not hold. For uniqueness, consider the example from Section 3: given 16 points in a square with Euclidean distance, the maximum diversity is achieved when two opposing vertices contain 8 objects each, and replacing any of these duplicates by any other object will not increase diversity. Continuity is trivial, as is the $O(n^2)$ complexity.

**Bottleneck and Energy**$(\gamma)$   Consider a collection of three objects, where $x_1$ and $x_2$ coincide, and $d_{13} = 1$. Increasing $d_{13}$ to $2$ will not change the diversity value, thus proving that monotonicity does not hold. Consider a collection of three coinciding objects. Replacing one of them with any other object does not change diversity value, thus proving that uniqueness does not hold. Continuity is trivial, as is the $O(n^2)$ complexity.

**SumBottleneck**   Consider a collection of four objects, where $x_1, x_2$ coincide, $x_3, x_4$ coincide, and $d_{13} = 1$. Increasing $d_{13} = d_{23} = d_{14} = d_{24}$ from $1$ to $2$ (while keeping $d_{12} = d_{34} = 0$) will not change diversity value, thus proving that monotonicity does not hold. Consider a collection of four objects, where $x_1, x_2, x_3$ coincide and $d_{14} = 10$. Replacing $x_3$ with a new object that has distance $1$ to $x_4$ will decrease diversity from $10$ to $2$, thus proving that uniqueness does not hold. Continuity is trivial, as is the $O(n^2)$ complexity.

**#Circles**$(t)$   Consider a collection of three objects with pairwise distances $4, 3, 2$. Increasing distance $3$ to $4$ will not change the diversity (for any $t$), thus proving that monotonicity does not hold. For a given $t$, consider a collection of two coinciding objects. Replacing the second of them with an object at distance $\frac{t}{10}$ from the first one does not change the diversity value, thus proving that uniqueness does not hold. The lack of continuity is trivial. Let us prove that the complexity of calculating #Circles$(t)$ is NP-hard. The problem of finding the size of the maximum complete subgraph (clique) in an unweighted undirected graph is known to be NP-hard. Consider any unweighted undirected graph $G$ with $n$ nodes. Construct a collection $X$ with $n$ objects corresponding to the nodes of $G$; the distance between two objects being $t$ if the corresponding nodes are connected and $0.9t$ otherwise. Suppose we computed #Circles$(t)$. Then, obviously, this value is also the size of the maximum clique in $G$. This proves that calculating #Circles$(t)$ is NP-hard.

**Unique**   Monotonicity, uniqueness, and continuity are trivial, as is the $O(n^2)$ complexity.

**HamDiv**   Consider a collection of four objects, where $d_{12} = d_{23} = d_{34} = d_{14} = 1$ and $d_{13} = d_{24} = 1.1$. Increasing $d_{13}$ from $1.1$ to $1.2$ will not change the diversity value, thus proving that monotonicity does not hold. Note that non-strict monotonicity holds; that is, if some pairwise distance is increased, the diversity value cannot decrease (although can stay the same). Now, consider two collections of four objects each. The objects are points on the Euclidean line. The first collection is: $0, 0, 1, 1$; the second collection is: $0, \frac{1}{3}, \frac{2}{3}, 1$. The diversity value for both collections is equal to $2$; thus, uniqueness does not hold. Continuity is trivial. Complexity is known to be NP-hard.

To prove the results for Vendi Score, DPP, and RKE, we first need to formulate the axioms in terms of similarities. Monotonicity requires that the measure monotonically increases when some of the pairwise similarities decrease. Uniqueness is formulated in terms of objects, and two objects being duplicates means that they have the maximum similarity value. Finally, continuity can be trivially reformulated.

**Vendi Score**   We first elaborate on the example of a violation of monotonicity from Section 3. Consider points on a circle with cosine similarity. Suppose the points $x_1, x_2, x_3$ are arranged in this order on the circle, the circle distance from $x_1$ to $x_2$ is $0.6$ radians, the distance from $x_2$ to $x_3$ is $1.4$ radians. Now we move $x_3$ by $0.1$ away from $x_1$ and $x_2$. Let us see what similarity matrices we have before and after this move:

$$S = \begin{pmatrix} 1 & \cos(0.6) & \cos(2.0) \\ \cos(0.6) & 1 & \cos(1.4) \\ \cos(2.0) & \cos(1.4) & 1 \end{pmatrix}, \quad \hat{S} = \begin{pmatrix} 1 & \cos(0.6) & \cos(2.1) \\ \cos(0.6) & 1 & \cos(1.5) \\ \cos(2.1) & \cos(1.5) & 1 \end{pmatrix}. \tag{3}$$

Vendi Score of $S$ is $1.941$ and Vendi Score of $\hat{S}$ is $1.916 < 1.941$, which is a violation of the monotonicity property.

Now suppose the points $x_1, x_2, x_3$ are arranged in this order on the circle, the circle distance from $x_1$ to $x_2$ is $0.2$ radians, the distance from $x_2$ to $x_3$ is $0.3$ radians. We replace $x_2$ by a duplicate of $x_1$. Let us see what similarity matrices we have

before and after this replacement:

$$S = \begin{pmatrix} 1 & \cos(0.2) & \cos(0.5) \\ \cos(0.2) & 1 & \cos(0.3) \\ \cos(0.5) & \cos(0.3) & 1 \end{pmatrix}, \quad \hat{S} = \begin{pmatrix} 1 & 1 & \cos(0.5) \\ 1 & 1 & \cos(0.5) \\ \cos(0.5) & \cos(0.5) & 1 \end{pmatrix}. \tag{4}$$

The corresponding collections of objects differ by replacing $x_2$ with a copy of $x_1$; that is, $S$ corresponds to $(x_1, x_2, x_3)$ and $\hat{S}$ corresponds to $(x_1, x_1, x_3)$. Vendi Score of $S$ is 1.187 and Vendi Score of $\hat{S}$ is $1.233 > 1.187$, which is a violation of the uniqueness property.

Continuity holds since $\exp\left(-\sum_{i=1}^{n} \lambda_i \log(\lambda_i)\right)$ continuously depends on $\lambda_1, \ldots, \lambda_n$, which in turn continuously depend on the similarity matrix. It is known that the complexity of finding the eigenvalues of a general (positive-semidefinite) matrix is $O(n^3)$, thus the complexity of calculating Vendi Score is also $O(n^3)$.

**DPP**   The example of a violation of monotonicity is shown in Section 3. To obtain the matrix $S$, we can consider three points $A$, $B$, $C$ on a unit 2D sphere with pairwise spherical distances between $A$ and $B$ equal to $\arccos(0.6) = 0.927$, between $B$ and $C$ equal to $\arccos(0.7) = 0.795$ and between $A$ and $C$ equal to $\arccos(0.2) = 1.369$. The similarity is given by the cosine function. For the matrix $\hat{S}$, we decrease the distance between $A$ and $C$ from $\arccos(0.2) = 1.369$ to $\arccos(0.3) = 1.266$, while keeping the distance between $A$ and $B$ unchanged, and the distance between $B$ and $C$ unchanged.

To prove that uniqueness is violated, consider a collection of three coinciding objects. Replacing one of them with any other object does not change the diversity value, thus proving that uniqueness is violated. Continuity is trivial. It is known that the complexity of finding the determinant of a general (positive-semidefinite) matrix is $O(n^3)$, thus the complexity of calculating $\det(S)$ is also $O(n^3)$.

**RKE**   Monotonicity, continuity, and complexity $O(n^2)$ are trivial. To prove that uniqueness is violated, we elaborate on the example from Section 3. Consider points on a circle with cosine similarity. Suppose the points $x_1, x_2, x_3$ are arranged in this order on the circle; the distance from $x_1$ to $x_2$ is 1.1 radians, the distance from $x_2$ to $x_3$ is 0.4 radians. Now, we replace $x_2$ with a duplicate of $x_3$. We get the following similarity matrices before and after the modification:

$$S = \begin{pmatrix} 1 & \cos(1.1) & \cos(1.5) \\ \cos(1.1) & 1 & \cos(0.4) \\ \cos(1.5) & \cos(0.4) & 1 \end{pmatrix}, \quad \hat{S} = \begin{pmatrix} 1 & \cos(1.5) & \cos(1.5) \\ \cos(1.5) & 1 & 1 \\ \cos(1.5) & 1 & 1 \end{pmatrix}. \tag{5}$$

The corresponding collections of objects differ by replacing $x_2$ with a copy of $x_3$; that is, $S$ corresponds to $(x_1, x_2, x_3)$ and $\hat{S}$ corresponds to $(x_1, x_3, x_3)$. RKE of $S$ is 0.564 and RKE of $\hat{S}$ is $0.584 > 0.564$, which is a violation of the uniqueness property.

**Species**$(q)$   Continuity and complexity $O(n^2)$ are trivial. Monotonicity is trivial for both cases $0 \le q < 1$ and $q > 1$. For violation of uniqueness, we consider the same example as for RKE. We computed Species$(q)$ of the collection $x_1, x_2, x_3$ and the collection $x_1, x_3, x_3$ for all $q$ in the range $[0, 100]$ with step size 0.001 (excluding $q = 1$ when Species$(q)$ is not defined). For all the considered $q$, the first collection has a lower value of Species$(q)$ than the second collection, which is a violation of the uniqueness property.

## C. Properties of MultiDimVolume

Let us prove that MultiDimVolume has monotonicity, uniqueness, continuity and is NP-hard to compute.

For convenience, we repeat the definition of MultiDimVolume. For a given $k$, $2 \le k \le n$, and a given submultiset $K$ of size $k$ of the multiset $X = \{x_1, \ldots, x_n\}$, calculate the product of all pairwise distances between the elements of $K$. Note that this product equals zero if at least two elements of $K$ coincide. Then, for a given $k$, we take the maximum of such products over all submultisets of size $k$ of $X$ and denote this maximum as $m_k(X)$. We define the diversity of $X$ as $\sum_{k=2}^{n} m_k(X)$.

Putting the above into one formula, we get:

$$\text{Diversity}(X) := \sum_{k=2}^{n} \max_{\substack{K \subseteq X \\ |K|=k}} \left( \prod_{\substack{x_i, x_j \in K \\ i<j}} d_{ij} \right). \tag{6}$$

Assume that we are given any distance matrix $D$ (or, equivalently, a collection of objects $X$). Denote by $\bar{k}$ the maximum $k$ such that $m_k(X)$ is non-zero. Note that by construction, $X$ includes exactly $\bar{k}$ pairwise non-coinciding objects and $m_{\bar{k}}(X)$ is the product of pairwise distances between these objects.

**Monotonicity**   We want to prove that MultiDimVolume is strictly monotone in $D$. Suppose we increase the distance between two objects $x_i$ and $x_j$ by $\epsilon > 0$; that is, we replace $d_{ij}$ by $d_{ij} + \epsilon$. Obviously, for every $k$, the value of $m_k(X)$ has not decreased. Thus, to prove monotonicity, it is sufficient to prove that at least one of $m_k(X)$ has increased. If $x_i$ and $x_j$ did not coincide before increasing $d_{ij}$, then after increasing $d_{ij}$ the term $m_{\bar{k}}(X)$ has increased since $d_{ij}$ is one of the multipliers in $m_{\bar{k}}(X)$. If $x_i$ and $x_j$ coincided before increasing $d_{ij}$, then after increasing $d_{ij}$ the collection $X$ includes exactly $\bar{k} + 1$ non-coinciding objects, and $m_{\bar{k}+1}(X)$ has increased from 0 to some positive value.

Note that for some matrices $D$ we cannot increase only one distance. For instance, if the objects $x_1, x_2, x_3$ coincide and we increase $d_{12}$ by $\epsilon$, we also need to simultaneously increase $d_{13}$ or $d_{23}$, otherwise we have $d_{13} = d_{23} = 0, d_{12} > 0$, which implies that $x_1$ coincides with $x_3$ and $x_2$ coincides with $x_3$, but $x_1$ and $x_2$ do not coincide. Clearly, the proof above easily generalizes to the case when we increase several distances simultaneously.

**Uniqueness**   Suppose $X$ includes at least one duplicate. We replace this duplicate with some new object that was not present in $X$. Then, $m_{\bar{k}+1}(X)$ has increased from 0 to some positive value. Also, for any $k \leq \bar{k}$, the values of $m_k(X)$ have not decreased. Thus, $\text{Diversity}(X)$ has increased.

**Continuity**   Note that MultiDimVolume is a composition of product, maximum, and sum that are all continuous functions. A composition of continuous functions is continuous. Thus, MultiDimVolume is continuous.

**NP-hardness**   Let us first prove that finding $m_k(X)$ for all $k$ is NP-hard. The problem of finding the size of the maximum complete subgraph (clique) in an unweighted undirected graph is known to be NP-hard. Consider any unweighted undirected graph $G$ with $n$ nodes. Construct a collection $X$ with $n$ objects corresponding to the nodes of $G$, the distance between two objects being 3 if the corresponding nodes are connected and 2 otherwise. Suppose we computed $m_k(X)$ for all $k$. Take the maximum $k$ such that $m_k(X) = 3^{\frac{k(k-1)}{2}}$. Then $k$ is the size of the maximum clique in $G$, which concludes the proof.

Although we proved that finding $m_k(X)$ for all $k$ is NP-hard, it does not directly imply that computing MultiDimVolume is NP-hard. Indeed, maybe we can compute MultiDimVolume without directly computing $m_k(X)$ for all $k$. Let us give a sketch of how to avoid this technical obstacle.

As above, consider a graph $G$ for which we want to find the size of the maximum clique. Construct a collection $X$ with $n$ objects corresponding to the nodes of $G$, the distance between two objects being $2 + \epsilon$ if the corresponding nodes are connected and 2 otherwise, where $\epsilon > 0$ is a small number (we will specify later how small it should be). Consider $m_k(X)$ for some $k$. It is the product of pairwise distances between some $k$ objects of $X$. Denote by $r_k$, $0 \leq r_k \leq \frac{k(k-1)}{2}$, the number of their pairwise distances which are equal to $2 + \epsilon$ (so, the remaining $\frac{k(k-1)}{2} - r_k$ distances are equal to 2). This is equivalent to saying that:

$$m_k(X) = (2 + \epsilon)^{r_k} 2^{\frac{k(k-1)}{2} - r_k} = 2^{\frac{k(k-1)}{2}} + \epsilon r_k 2^{\frac{k(k-1)}{2} - 1} + O\left(\epsilon^2\right).$$

Therefore,

$$\text{Diversity}(X) = \sum_{k=2}^{n} m_k(X) = \left( \sum_{k=2}^{n} 2^{\frac{k(k-1)}{2}} \right) + \epsilon \sum_{k=2}^{n} r_k 2^{\frac{k(k-1)}{2} - 1} + O\left(\epsilon^2\right).$$

Note that for a given $n$, the value of $\epsilon$ can be chosen sufficiently small so that the last term $O(\epsilon^2)$ is negligibly small compared to the other two terms.

Now suppose we know $\text{Diversity}(X)$. We also know the term $\sum\limits_{k=2}^{n} 2^{\frac{k(k-1)}{2}}$ and we know $\epsilon$. Thus, we can compute $\sum\limits_{k=2}^{n} r_k 2^{\frac{k(k-1)}{2}-1}$.

We claim that knowing the value $M = \sum\limits_{k=2}^{n} r_k 2^{\frac{k(k-1)}{2}-1}$ we can recover $r_2, r_3, \ldots, r_n$. For this, we note that for any $k = 3, \ldots, n$:

$$2^{\frac{k(k-1)}{2}-1} > \sum_{i=2}^{k-1} \frac{i(i-1)}{2} \cdot 2^{\frac{i(i-1)}{2}-1}. \tag{7}$$

Indeed, this holds for $k = 3$ and it is easy to check that the left-hand side of the inequality grows faster than the right-hand side.

Now, consider $k = n$ and note that the left-hand side of (7) is equal to how much the value of $M$ changes if we change $r_n$ by 1. In turn, the right-hand side of (7) is an upper bound on the sum of all other terms in $M$. Thus, knowing $M$, we can find the maximum integer $r_n$ such that $r_n 2^{\frac{n(n-1)}{2}-1} \le M$. After we found $r_n$, we get rid of the term $r_n 2^{\frac{n(n-1)}{2}-1}$ and can do the same reasoning to find $r_{n-1}$, and continue until we find all $r_2, \ldots, r_n$. After that, we take the maximum $k$ such that $r_k = \frac{k(k-1)}{2}$, this $k$ is the size of the maximum clique in $G$, which concludes the proof.

## D. Properties of IntegralMaxClique

Let us prove that IntegralMaxClique has monotonicity, uniqueness, continuity and is NP-hard to compute.

For convenience, we repeat the definition of IntegralMaxClique. For a given threshold $t \ge 0$, we construct the following graph. The nodes are $x_1, \ldots, x_n$. Two nodes $x_i$ and $x_j$ are connected by an edge iff $d_{ij} \ge t$, and we assign $d_{ij}$ as the weight of this edge. We find a clique (complete subgraph) in this graph with the maximum number of nodes. If there are several such cliques, we pick the one with the maximum total weight of edges. For the chosen clique, we calculate the total weight of its edges and denote it by $w_t(X)$. Then, we define diversity as

$$\text{Diversity}(X) := \int_0^{+\infty} w_t(X)\, dt. \tag{8}$$

Assume that we are given any distance matrix $D$ (or, equivalently, a collection of objects $X$). Denote by $\bar{d}$ the smallest non-zero pairwise distance between the objects of $X$. If all pairwise distances are 0, then monotonicity is trivial, so we can assume $\bar{d} > 0$. Note that for $t \le \bar{d}$, the value of $w_t(X)$ is the sum of pairwise distances between all pairwise non-coinciding elements of $X$.

**Monotonicity** Assume that we increase the distance between two objects $x_i$ and $x_j$ by $\epsilon > 0$, that is, we replace $d_{ij}$ by $d_{ij} + \epsilon$. Obviously, for every $t$, the value of $w_t(X)$ has not decreased. If $d_{ij} > 0$, then for all $t \le \bar{d}$ the term $d_{ij}$ is a summand in $w_t(X)$, thus for every $t \le \bar{d}$ the value of $w_t(X)$ has increased at least by $\epsilon$. Therefore, $\text{Diversity}(X)$ has increased by at least $\bar{d}\epsilon$. If $d_{ij} = 0$, then for all $t \le \epsilon$, the value of $w_t(X)$ has increased by at least $\epsilon$ (since a new element is added to the maximum clique). Thus, $\text{Diversity}(X)$ has increased by at least $\epsilon^2$.

As with MultiDimVolume, this proof can be easily generalized to the case where several distances are increased simultaneously.

**Uniqueness** Suppose $X$ includes at least one duplicate. We remove one duplicated element and add some new object that was not present in $X$. Suppose the distance from the new object to the nearest object from $X$ is $r > 0$. Then, for $t \le r$, the value of $w_t(X)$ has increased by at least $r$, and for every $t > r$, the value of $w_t(X)$ has not decreased. Thus, $\text{Diversity}(X)$ has increased by at least $r^2$. This shows that making duplicates distinct increases diversity, establishing uniqueness.

**Continuity** Assume that we increase the distance between two objects $x_i$ and $x_j$ by $\epsilon > 0$, that is, we replace $d_{ij}$ by $d_{ij} + \epsilon$. Let us see how much $\text{Diversity}(X)$ could change. Obviously, for every $t$, the value of $w_t(X)$ has not decreased.

Let us estimate how much $\text{Diversity}(X)$ could increase. We decompose the integral into three parts:

$$\text{Diversity}(X) := \int_0^{+\infty} w_t(X)\,dt = \int_0^{d_{ij}} w_t(X)\,dt + \int_{d_{ij}}^{d_{ij}+\epsilon} w_t(X)\,dt + \int_{d_{ij}+\epsilon}^{+\infty} w_t(X)\,dt. \tag{9}$$

It is easy to prove that for $t \le d_{ij}$, the value of $w_t(X)$ could increase at most by $\epsilon$, thus the first part could increase at most by $\epsilon d_{ij}$ (since we integrate from 0 to $d_{ij}$). For the second term, we note that $w_t(X)$ is bounded from above by $\left( \sum_{k<l} d_{kl} \right) + \epsilon$, thus the second term is bounded by $\epsilon \left( \sum_{k<l} d_{kl} \right) + \epsilon^2$ and could increase by at most this value. The third term does not change since for $t > d_{ij} + \epsilon$ the value of $w_t(X)$ does not change.

Therefore, $\text{Diversity}(X)$ has increased by at most $\epsilon d_{ij} + \epsilon \left( \sum_{k<l} d_{kl} \right) + \epsilon^2$. So, if we increase $d_{ij}$ by $\epsilon$, then $\text{Diversity}(X)$ increases by at most $\epsilon c$, where $c$ is some constant independent of $\epsilon$ (given that $\epsilon < 1$, so the term $\epsilon^2$ is bounded by $\epsilon$). From this, the continuity follows.

**NP-hardness**    The problem of finding the size of the maximum complete subgraph (clique) in an unweighted undirected graph is known to be NP-hard. Consider any unweighted undirected graph $G$ with $n$ nodes. Construct a collection $X$ with $n$ objects corresponding to the nodes of $G$, the distance between two objects being 3 if the corresponding nodes are connected and 2 otherwise. Suppose we computed $\text{Diversity}(X)$. Let us show how to find the size of the maximum clique in $G$. Note that for $t \le 2$, the value of $w_t(X)$ is the sum of all pairwise distances in $X$, that is, $\sum_{k<l} d_{kl}$ (which can be computed in $O(n^2)$ time). For $2 < t \le 3$, the value of $w_t(X)$ is $3\frac{s(s-1)}{2}$, where $s$ is the size of the maximum clique in $G$. For $t > 3$, the value of $w_t(X)$ is 0. So, we get $\text{Diversity}(X) = 2 \sum_{k<l} d_{kl} + 3\frac{s(s-1)}{2}$, from which we can find $s$ in constant time. Thus, once we know $\text{Diversity}(X)$, we can find $s$ in $O(n^2)$ time. This proves that calculating $\text{Diversity}(X)$ is NP-hard.

## E. Proofs of Propositions from Section 5

### E.1. Proof of Proposition 6.1

Let us first recall the statement of the proposition. Suppose a diversity function has uniqueness and continuity. Let $x_1, \ldots, x_k$ be a set of $k$ pairwise distinct objects. Let $C$ be a multiset of $n - k$ objects, each of which coincides with one of $x_1, \ldots, x_k$. Then, diversity of the multiset $\{x_1, \ldots, x_k\} \cup C$ is the same for all such $C$.

Consider the following lemma.

**Lemma E.1.** *Suppose a diversity function has uniqueness and continuity. Let $x_1, \ldots, x_{n-1}$ be any collection of $n - 1$ objects. We denote by $A_1$ the collection of $n$ objects $x_1, \ldots, x_{n-1}, x_1$ and by $A_2$ the collection of $n$ objects $x_1, \ldots, x_{n-1}, x_2$ (note that $A_1$ and $A_2$ differ only by the last object). Then, $\text{Diversity}(A_1) = \text{Diversity}(A_2)$.*

Informally, this lemma says that we can remove the duplicate of $x_1$ and add the duplicate of $x_2$ without changing the value of the diversity function. The proposition trivially follows from this lemma, so it is sufficient to prove it.

W.l.o.g., assume that $\text{Diversity}(A_1) - \text{Diversity}(A_2) = \epsilon > 0$. Denote by $A_2'$ the following collection: take $A_2$ and increase the distance from the last object to all objects by a small $\delta > 0$ in such a way that diversity changes by less than $\frac{\epsilon}{2}$ (note that the last object is no longer a duplicate). By continuity, this is possible. Then, $\text{Diversity}(A_2')$ is less than $\text{Diversity}(A_2) + \frac{\epsilon}{2}$. Thus, $\text{Diversity}(A_2') < \text{Diversity}(A_1)$. However, by uniqueness, we have $\text{Diversity}(A_2') > \text{Diversity}(A_1)$ since the last object of $A_2'$ is not a duplicate, and the last object of $A_1$ is a duplicate. So, we get a contradiction which concludes the proof of the lemma.

### E.2. Proof of Proposition 6.2

We need to prove that a diversity function satisfying all the axioms cannot be decomposed in the following form:

$$\text{Diversity}(X) = F(d_{12}, d_{13}, \ldots, d_{1n}) + G(x_2, \ldots, x_n).$$

Suppose we increase $d_{12}$ (and $d_{21}$) by some $\Delta$. Then, diversity will increase by the following value:

$$F(d_{12} + \Delta, d_{13}, \ldots, d_{1n}) + G(x_2, \ldots, x_n) - F(d_{12}, d_{13}, \ldots, d_{1n}) - G(x_2, \ldots, x_n) =$$
$$= F(d_{12} + \Delta, d_{13}, \ldots, d_{1n}) - F(d_{12}, d_{13}, \ldots, d_{1n}). \quad (10)$$

Note that by permutation invariance we can decompose $\mathrm{Diversity}(X)$ based not on $x_1$, but on $x_2$:

$$\mathrm{Diversity}(X) = F(d_{21}, d_{23}, \ldots, d_{2n}) + G(x_1, x_3, \ldots, x_n). \quad (11)$$

Using this decomposition, we see that when we increase $d_{12}$ (and $d_{21}$) by $\Delta$, the diversity increases by the following value:

$$F(d_{21} + \Delta, d_{23}, \ldots, d_{2n}) + G(x_1, x_3, \ldots, x_n) - F(d_{21}, d_{23}, \ldots, d_{2n}) - G(x_1, x_3, \ldots, x_n) =$$
$$= F(d_{21} + \Delta, d_{23}, \ldots, d_{2n}) - F(d_{21}, d_{23}, \ldots, d_{2n}) \quad (12)$$

Combining the results of (11) and (12), we get:

$$F(d_{12} + \Delta, d_{13}, \ldots, d_{1n}) - F(d_{12}, d_{13}, \ldots, d_{1n}) = F(d_{21} + \Delta, d_{23}, \ldots, d_{2n}) - F(d_{21}, d_{23}, \ldots, d_{2n}).$$

Note that the left part depends on $d_{13}, \ldots, d_{1n}$, while the right part does not depend on these variables. Similarly, the right part depends on $d_{23}, \ldots, d_{2n}$, while the left part does not depend on these variables. This means that both parts actually do not depend on any of $d_{13}, \ldots, d_{1n}$ and $d_{23}, \ldots, d_{2n}$, so they depend only on $d_{12}$ (or $d_{21}$, which is the same) and $\Delta$. Thus, we proved that if we increase $d_{12}$ by $\Delta$, the diversity changes by some value that depends only on $d_{12}$ and $\Delta$ and does not depend on other pairwise distances. By permutation invariance, for any $d_{ij}$ the analogous statement is true. From these statements, it easily follows that

$$\mathrm{Diversity}(X) = h(d_{12}) + h(d_{13}) + \ldots = \sum_{i<j} h(d_{ij}),$$

where we have the same function $h$ applied to all distances by permutation invariance.

Consider the following collection: the first $n - 1$ objects are duplicates of one element, and the last object is at distance 1 from them. So, there are $n - 1$ pairwise distances of 1 and $\frac{(n-1)(n-2)}{2}$ distances of 0. Thus, diversity is $(n - 1)h(1) + \frac{(n-1)(n-2)}{2}h(0)$. Using Proposition 6.1, we can move one of the duplicates in such a way that now it duplicates the last object, and diversity should not change. Now, there are $2(n - 2)$ pairwise distances of 1, and $\frac{(n-2)(n-3)}{2} + 1$ distances of 0. Thus, diversity is $2(n - 2)h(1) + \left( \frac{(n-2)(n-3)}{2} + 1 \right) h(0)$. So, we get

$$(n - 1)h(1) + \frac{(n - 1)(n - 2)}{2}h(0) = 2(n - 2)h(1) + \left( \frac{(n - 2)(n - 3)}{2} + 1 \right) h(0),$$

from which we get $(n - 3)h(1) = (n - 3)h(0)$, which implies $h(1) = h(0)$ (given that $n > 3$). Monotonicity implies that $h$ is strictly monotone, which contradicts $h(1) = h(0)$, which concludes the proof.

