# OpenReview forum: "Measuring Diversity: Axioms and Challenges"
_ICML.cc/2025/Conference — ICML 2025 poster_

### Official Review · Reviewer_Huhr · 2025-03-12

**Overall Recommendation:** 4

**Summary:**

This paper examines in depth the problem of quantifying diversity for a set of objects, a concept widely used in various fields such as image generation, molecule generation, and recommendation systems. The authors conduct a systematic review of existing diversity measures and highlight their undesirable behaviors in certain cases. Based on this analysis, they formulate three desirable properties (axioms) for a reliable diversity measure: monotonicity, uniqueness, and continuity. The paper demonstrates that none of the existing measures simultaneously satisfy these three axioms, thus suggesting their inadequacy for a rigorous quantification of diversity. Subsequently, the authors construct two examples of measures that possess all the desirable properties, proving that the set of axioms is not self-contradictory. However, these constructed examples turn out to be computationally too complex for practical use, leading to an open problem: that of designing a diversity measure that satisfies all the axioms while being efficiently computable.

# update after rebuttal
No need to change my score.

**Claims And Evidence:**

This is a purely theoretical paper, which can have a definitive impact on the practice. All the claims are thoroughly either illustrated or demonstrated.

The paper is well written and richly illustrated.

Although the paper is theoretical and contains no experiments, I like it, and I believe it shall be published.

**Essential References Not Discussed:**

None

**Experimental Designs Or Analyses:**

There is no experimental evaluation per se. However, the examples illustrating the different claims are convincing.

**Methods And Evaluation Criteria:**

There is no application, this is a purely theoretical paper.

**Other Comments Or Suggestions:**

--

**Other Strengths And Weaknesses:**

--

**Questions For Authors:**

--

**Relation To Broader Scientific Literature:**

The interest of the paper is to provide insights into measures of diversity that are frequently used in practice. As none of these measures satisfy the desired properties, it is important for the end-user to understand why and when they are failing, so that the measures can be used properly.

Furthermore, the question of having an usable measure that satisfy the three desirable properties, which is an open problem in the paper, is of great interest.

**Theoretical Claims:**

As far as I have been able to check, the proofs are correct.

---

> ### Author Rebuttal · Authors · 2025-04-01
>
> Thank you for your thoughtful review and support! We sincerely appreciate that you acknowledge theoretical soundness of our research and its relevance to practice.

---

### Official Review · Reviewer_zmT9 · 2025-03-13

**Overall Recommendation:** 3

**Summary:**

This paper studies the problem of diversity measurement and proposes three axioms—monotonicity, uniqueness, and continuity—as necessary conditions for a reliable diversity measure. The authors analyze existing diversity measures and demonstrate that none satisfy all three axioms. To address this gap, the paper constructs two theoretical diversity measures that adhere to the proposed axioms but are computationally infeasible. The work highlights an open challenge: finding a computationally efficient diversity measure that aligns with the theoretical principles.

**Claims And Evidence:**

The paper presents the following key claims:
1.The three proposed axioms (monotonicity, uniqueness, and continuity) establish a theoretical basis for evaluating diversity measures.
2.The paper systematically reviews existing diversity measures and demonstrates that they fail to satisfy all three axioms.
3.Two new diversity measures (MultiDimVolume and IntegralMaxClique) are proposed that satisfy the axioms but have NP-hard complexity.
The claims are well-supported by rigorous theoretical analysis, but the lack of experimental validation weakens their practical impact.

**Essential References Not Discussed:**

The paper does not reference works on diversity in active learning, dataset selection, or representation learning.

**Experimental Designs Or Analyses:**

No experiments are provided to validate the proposed axioms.
The paper does not test existing diversity measures on real-world datasets.
The computational feasibility of the new diversity measures is not explored through empirical benchmarks.

**Methods And Evaluation Criteria:**

The axiomatic approach provides a novel and well-structured evaluation framework.The systematic review of existing methods is insightful and identifies key limitations.
However, the absence of benchmark datasets and experimental results limits the validation of the proposed axioms in real-world applications.
The evaluation criteria focus solely on theoretical correctness, neglecting computational efficiency and practical applicability.

**Other Comments Or Suggestions:**

Please see the above weakness

**Other Strengths And Weaknesses:**

Strengths
1.The paper introduces a novel axiomatic framework for diversity measurement, which provides a theoretical foundation for evaluating and comparing different diversity measures.
2.The paper provides well-structured proofs to demonstrate that existing diversity measures fail to satisfy all three proposed axioms. The authors also construct two diversity measures that adhere to these axioms, proving their feasibility in a theoretical context.
Weaknesses
1.While the paper proposes diversity measures that satisfy the axioms, they are computationally infeasible (NP-hard), limiting their applicability in real-world scenarios. The paper does not explore approximations or alternative methods that balance theoretical soundness with practical efficiency.
2.The paper does not provide experimental results to demonstrate the impact of the proposed axioms on real-world datasets. The theoretical framework is well-developed, but its practical implications remain unclear without empirical evaluation.
3.The paper does not sufficiently explore how these axioms might be relevant to practical machine learning tasks, such as dataset selection, diversity-aware training, or clustering evaluation.
4.The proposed framework assumes that all pairwise distances are equally important, neglecting the possibility that hierarchical structures or different levels of diversity may be relevant in real-world applications. Considering weighted distances or context-aware diversity measures could improve the framework’s applicability.

**Questions For Authors:**

Please see the above weakness

**Relation To Broader Scientific Literature:**

The paper builds upon existing work on diversity measurement but does not discuss its implications for real-world applications such as active learning, data augmentation, or clustering
Author should discuss potential applications of the proposed framework in real-world machine learning tasks.

**Theoretical Claims:**

The theoretical claims are well-structured and logically sound. And proofs are detailed and rigorously support the proposed axioms.
However, no evidence is provided to show whether satisfying these axioms leads to improved diversity measurement in practical scenarios.

---

> ### Author Rebuttal · Authors · 2025-04-01
>
> Thank you for your suggestions and positive feedback! We appreciate that you find our approach novel, well-structured and theoretically sound. We address the raised concerns below.
>
> > The paper does not explore approximations or alternative methods that balance theoretical soundness with practical efficiency.
>
> In the paper, we briefly discuss that some measures can still be suitable in some applications despite not satisfying all the properties. E.g., in Section 3, we discuss undesirable behavior w.r.t. optimization and note that not all measures have this problem. In lines 204-211, we mention that Bottleneck, SumBottleneck and Energy behave well w.r.t. optimization and thus can be used as target measures when constructing a diverse dataset. Energy has all the desirable properties when all elements are unique. When some elements coincide, this measure becomes infinite and thus the properties are not satisfied, which makes Energy inappropriate for comparing diversity of different datasets. On the other hand, while we do not advise optimizing Average, its value is interpretable and can be used as one of the measures evaluating dataset diversity.
>
> > The paper does not test existing diversity measures on real-world datasets.
>
> To demonstrate that the choice of diversity measure is important and to illustrate shortcomings of a particular measure, we conducted the following experiment. We consider the setup of generating structurally diverse graphs from Velikonivtsev et al. (2024). Using the code publicly shared by the authors, we compare diverse graphs generated by a genetic algorithm while optimizing either Energy (as in the original paper) or Average. The obtained results can be found [here](https://anonymous.4open.science/r/ICML_2025_rebuttal-3966). We see that optimizing Average (portrait_genetic_optimizing_avg.pdf) leads to more similar graph structures that tend to be either too dense or too sparse, which agrees with our observations about corner cases for Average in Section 2. In turn, Energy is suitable for optimization and thus leads to more structurally diverse graphs.
>
> > No experiments are provided to validate the proposed axioms.
>
> Validating axioms is challenging: one can potentially compare two measures, one of which satisfies a particular axiom while the second one does not, but the result will not allow one to make conclusions about the axiom since there can be other properties affecting the comparison of the measures. That is why we first analyzed different measures and observed their failure cases and then formulated properties that are intuitively desirable for all diversity measures. If one knows which properties a particular measure does or does not satisfy, one can better interpret the obtained results.
>
> > Author should discuss potential applications of the proposed framework in real-world machine learning tasks.
>
> Thank you for your suggestion. We will include a deeper discussion on how diversity measurement is used in machine learning tasks. Specifically, we will highlight that a well-defined diversity measure ensures informative sample selection in active learning, guides the generation of meaningful synthetic data in augmentation, and provides a way to assess cluster distinctiveness.
>
> > The appendix provides extensive proofs but lacks practical implementation details.
>
> Note that the main contribution of our paper is theoretical and the paper does not contain experiments. However, if there are any questions regarding potential implementations or applications, we will be happy to address them.
>
> > The evaluation criteria focus solely on theoretical correctness, neglecting computational efficiency and practical applicability.
>
> Let us remark that in the paper we pay attention to the computational efficiency of diversity measures: asymptotic computational complexity is analyzed and reported in Table 2. Both MultiDimVolume and IntegralMaxClique are NP-hard which is why we pose an important open problem on whether there exists a more efficient measure that satisfies all the properties.
>
> > The computational feasibility of the new diversity measures is not explored through empirical benchmarks.
>
> Let us note that the main goal of constructing these two measures was to show that our set of axioms is not self-contradictory. Since these measures are NP-hard, they cannot be used in most applications. That is why we believe that the open problem that we pose is important and hope that it will be addressed in future studies.
>
> > Considering weighted distances or context-aware diversity measures could improve the framework’s applicability.
>
> Thank you for this suggestion! We believe that studying more complex scenarios is an important direction for future research. In our paper, we address the simplest case and observe that it is already challenging.
>
> We are open to further discussions!

---

### Official Review · Reviewer_Pyqh · 2025-03-14

**Overall Recommendation:** 4

**Summary:**

This paper explores how to quantify diversity for a set of objects. The authors first review existing diversity measures, showing that they can have undesirable behaviors. To address this, the paper suggests three properties that a diversity measure should have: monotonicity (diversity should increase as pairwise distances between objects increase), uniqueness (replacing an object with a duplicate should decrease diversity), and continuity (diversity should be a continuous function of pairwise distances).

The paper shows that none of the existing measures satisfies all three properties.  The authors provide examples of measures that do satisfy the properties, but these are complex for practical use. The paper concludes by posing the open problem of finding a measure that has all three properties and is computationally feasible.

**Claims And Evidence:**

The authors support their claim that existing diversity measures exhibit undesirable behavior by providing examples and visual representations in Section 3.

In Section 4, the authors clearly define the axioms and provide justifications for why these axioms are important for a reliable diversity measure. The proofs for the properties of existing measures and the proposed measures are detailed and can be found in the appendices.

One potential issue I think is with Axiom 3 (Continuity). A diversity function must be continuous. The authors claim that the function should naturally be continuous, and provide a counter example in Appendix. In this example, I do not agree the point that `the right configuration is intuitively more diverse`. The diversity in this case depends on the distance measure. For example, even with continuous functions, if you have very high penalty on duplicates (and/or near duplicates), you will still get such behavior, so the problem is not with continuity.

**Essential References Not Discussed:**

There has been a trend in methods that use human feedback/judgment to learn a high-dimensional distance measure [1] or abstract diversity metrics [2]. Given that the authors touched human intuition on diversity, a discussion on these related work would be helpful to understand how the axioms and challenges apply to more practical settings such as GenAI.

[1] Fu, Stephanie, et al. "DreamSim: Learning New Dimensions of Human Visual Similarity using Synthetic Data." Advances in Neural Information Processing Systems 36 (2023): 50742-50768.
[2] Ding, Li, et al. "Quality diversity through human feedback: towards open-ended diversity-driven optimization." Proceedings of the 41st International Conference on Machine Learning. 2024.

**Experimental Designs Or Analyses:**

N/A

**Methods And Evaluation Criteria:**

N/A

**Other Comments Or Suggestions:**

N/A

**Other Strengths And Weaknesses:**

N/A

**Questions For Authors:**

See Claims And Evidence.

**Relation To Broader Scientific Literature:**

The paper makes significant contributions to the scientific literature by analyzing existing diversity measures, proposing a new set of axioms, identifying a gap in the literature, and proposing new measures that adhere to the axioms. The authors also clearly position their work within the context of prior research and highlight the direction for future work.

**Theoretical Claims:**

The claims look good to me except the one with continuity mentioned above.

---

> ### Author Rebuttal · Authors · 2025-04-01
>
> Thank you for your thoughtful comments and positive feedback!
>
> We see that the main concern is regarding the necessity of the continuity axiom, so let us elaborate on this subject.
>
> **Continuity axiom**
>
> First, it is natural to assume that minor changes in object locations should lead to small changes in diversity: this is important for interpretable comparison of diversity values across datasets. Second, we believe that this property is critical since without assuming it one may come up with a range of measures that do satisfy the remaining two properties while being intuitively not useful for measuring diversity. The reasoning below extends the example in Appendix A.
>
> Consider any measure $M$ that is monotone (e.g., Average). Apply any order-preserving transformation to $M$ such that the range of the resulting measure $M’$ is in $[0,1)$. E.g., if $M$ takes only non-negative values, we can take $M’(X):=1-e^{M(X)}$. Then, the measure $Unique(X) + M’(X)$ has both uniqueness and monotonicity. Essentially, $Unique(X) + M’(X)$ compares two configurations of points in the following way:
>
> - Count the number of unique points in both configurations;
> - If the numbers of unique points are different, then the configuration with a bigger number is more diverse;
> - If the numbers of unique points are the same, compare configurations based on the measure $M$ (or, equivalently, $M’$).
>
> We argue that $Unique(X) + M’(X)$ is (in many cases) not a good diversity measure. For this, consider any measure $M$ that has the monotonicity property (for instance Average), optimize it, and after that spread the points a bit to make them unique. Then, we get the (nearly) optimal configuration for $Unique(X) + M’(X)$ which is very similar to the optimal configuration for $M(X)$. However, the optimal solution for $M(X)$ may be not diverse, as we show with the example for Average on page 3.
>
> The discreteness of $Unique(X)$ plays a crucial role in the construction above. A natural way to prevent the measures of the form $Unique(X) + M’(X)$ form being considered as a good diversity measure is to require that diversity measures must be continuous.
>
> We will add the above reasoning to Appendix A.
>
> Regarding the visual example in Appendix A, we note that the penalty of a duplicate is higher than any increase in diversity that can be gained from moving already unique points further away from each other. We claim that all continuous monotone diversity measures do not show such behavior. Let us formally prove this.
>
> The measure from Appendix A rate any configuration of $16$ unique points as more diverse than any configuration of $15$ unique points and $1$ duplicate. Suppose that a continuous measure $M$ does the same. For $a \ge 0$ denote by $X_a$ the set of $16$ points with pairwise distances $a.$ For $a > 0$ denote by $Y_a$ the set of $16$ points from which $15$ have pairwise distances $a$ and one point is a duplicate. Let $r = M(Y_2)-M(Y_1)$, note that $r>0$ by monotonicity of $M$. By continuity of $M$ we can find $1>\epsilon >0$ such that $(M(X_\epsilon) - M(X_0)) < r/10$ and $(M(Y_\epsilon) - M(X_0)) < r/10$. Then $M(X_\epsilon)$ and $M(Y_\epsilon)$ differ by at most $r/5.$ At the same time, $M(Y_1) > M(Y_\epsilon)$ (since $\epsilon<1$) and $M(Y_2)-M(Y_1) =r,$ thus $M(Y_2)$ is bigger than $M(X_\epsilon)$ by at least $4r/5>0$. This is a contradiction since we assumed that $M(X_a)>M(Y_b)$ for all $a>0, b>0.$
>
> > There has been a trend in methods that use human feedback/judgment to learn a high-dimensional distance measure [1] or abstract diversity metrics [2]. Given that the authors touched human intuition on diversity, a discussion on these related work would be helpful to understand how the axioms and challenges apply to more practical settings such as GenAI.
>
> Thank you for the references, we will extend the related work accordingly. The mentioned papers use human feedback/judgment to estimate pairwise distances between the objects. A good distance measure is a critical ingredient of a reliable diversity measure. In our work, we assume that pairwise distances are given and analyze different ways of aggregating these values into diversity.
>
> We hope that our response addresses your concerns. We are open to further discussions!

---

### Official Review · Reviewer_yDDf · 2025-03-16

**Overall Recommendation:** 2

**Summary:**

This paper discusses metrics for measuring diversity in various applications. The authors review existing diversity measures and highlight their limitations in corner cases. They propose three key properties—monotonicity, uniqueness, and continuity—that a reliable diversity measure should possess. The paper demonstrates that no existing measure satisfies all three properties, making them unsuitable for accurate diversity quantification. While the authors construct two examples of measures that meet these properties, they, too happen to be NP-Hard, and the paper concludes by posing the challenge of creating a feasible diversity measure.

## update after rebuttal

I've added follow-up comments for the authors, and I sincerely hope they'll resolve them in the next version of the manuscript. I have also raised score by a point.

**Claims And Evidence:**

Given the nature of this paper, the provided claims and evidence are clear. However, I think there are a few weakness

1. The paper is quite similar to Velikonivtsev et al. 2024, which also talks about the first two axioms and many of the mentioned previous works. Continuity being the new axiom, I am not fully convinced with the example that it is important (mentioned in Appendix A) in particular, the entire argument of diversity is using the intuitive notion. For instance, I might consider that the Fig on right in Appendix A less diverse because of even a single duplicate (recall that you're changing two things at a same time, spread and adding duplicate).
Hence is no reason to believe that this intuitive notion of diversity will result in practical improvement for whatever downstream task is considered.
2. Can authors cite and discuss about the following work - "Position: Measure Dataset Diversity, Don't Just Claim It". ICML'24
3. Axioms could've been explained formally using math. That is, this work should have defined the exact notation for the diversity function, what are the arguments (subspace of all matrices defined on field R, multi set of examples), how is monotonicity mathematically defined for this multivariable function.
4. This paper misses on connecting the works with the vast literature on submodular functions, which are known to capture diversity, and appear in many areas of science. See -- "Submodularity In Machine Learning and Artificial Intelligence" for a gentle introduction.
5. In most of the practical situations one is interested in coming up with a subset of the data starting from no datapoints. This is something I am not sure how the current framework tackles (a discussion is missing)

**Essential References Not Discussed:**

1. "Position: Measure Dataset Diversity, Don't Just Claim It". ICML'24
2. "Submodularity In Machine Learning and Artificial Intelligence"

**Experimental Designs Or Analyses:**

While 2D experiments are good to demonstrate, proposed method is not practical (even hard to compute, let alone optimize), therefore any empirical demonstration is not possible.

**Methods And Evaluation Criteria:**

1. From the point of view of picking out corner cases for existing measures, 2D examples are great. However, they still are toy examples at the core. Often, people are interested in finding a diverse subset of a given giant dataset; hence, do these corner cases (particularly for the determinant-based examples) occur regularly?
2. Unfortunately, the proposed method is also NP-hard. While the work says that it is quite difficult to satisfy all three axioms, it would've been nice to discuss an existing metric that doesn't succumb to corner cases in practical situations.

**Other Comments Or Suggestions:**

Refer to the claims and evidence

**Other Strengths And Weaknesses:**

Refer to the claims and evidence

**Questions For Authors:**

Refer to the claims and evidence

**Relation To Broader Scientific Literature:**

The work lacks discussion with the entire area of submodular functions, which are at the core of measuring data diversity.

**Theoretical Claims:**

I've gone through the proofs provided in the appendix.

---

> ### Author Rebuttal · Authors · 2025-04-01
>
> Thank you for the feedback and suggestions! We address the concerns below and will be happy to discuss any of the raised issues further.
>
> > Continuity being the new axiom, I am not fully convinced with the example that it is important (mentioned in Appendix A)
>
> Regarding the importance of the continuity axiom, we first note that it is natural to assume that minor changes in object locations should lead to small changes in diversity: this is important for interpretable comparison of diversity values across datasets. Also, please see our response to **Reviewer Pyqh (Continuity axiom).**
>
> Also, we modify the definitions of both Monotonicity and Uniqueness. This modification is important (see lines 244-250 of Section 4.2) and makes developing a suitable diversity measure more challenging.
>
> > Can authors cite and discuss about the following work - "Position: Measure Dataset Diversity, Don't Just Claim It". ICML'24
>
> Thank you for the reference. This paper gives motivation on why measuring diversity is important and argues that it is critical to provide a clear definition of diversity when analyzing datasets. We will cite this paper in the introduction.
>
> > Axioms could've been explained formally using math.
>
> In the paper, we opted for more intuitive definitions and will add formal versions in the revised text.
>
> As defined in the paper, let $D_n$ be a subset of all $n \times n$ matrices satisfying the properties in lines 247-251. There is a natural action of the symmetric group $S_n$ on this subset (which simultaneously rearranges rows and columns). Diversity function is any $S_n$-invariant function from $D_n$ to $\mathbb{R}$.
>
> **Axiom 1 (Monotonicity).** For any two matrices $A, B \in D_n$ such that $\forall i,j: a_{ij} \ge b_{ij}$ and at least one of inequalities is strict, we have $\mathrm{Diversity}(A) > \mathrm{Diversity}(B)$.
>
> **Axiom 2 (Uniqueness).** Suppose that for $A,B \in D_n$:
>
> - $a_{ij}=b_{ij}$ for all $i>2, j>2$
> - $b_{1i}=b_{2i}$ for all $i$
> - $a_{2j}=b_{2j}$ for all $j>2$
> - $a_{1j}>0$ for all $j>1$
>
> Then $\mathrm{Diversity}(A) > \mathrm{Diversity}(B)$.
>
> **Axiom 3  (Continuity).** The space $D_n$ is endowed with the subspace topology induced by the standard topology on the set of all $n \times n$ matrices. A diversity function must be continuous w.r.t. this topology.
>
> > This paper misses on connecting the works with the vast literature on submodular functions, which are known to capture diversity, and appear in many areas of science.
>
> Thank you for the reference, we will add a discussion on submodular functions in the updated version.
>
> Submodular functions indeed can be used to capture diversity and there is a condition that a submodular function should satisfy that captures an intuitive behavior of diversity/coverage as some elements are added to a dataset (note that in our work we consider properties that hold when the number of elements is fixed).
>
> Usually, such functions are used as a diversity measure of a set of objects $X$, when the set of all possible objects $V$ is known. Thus, some known submodular functions use summation (or integration) over the set $V$ (for instance, see the facility location function). Our setting is more general: we want to measure diversity of $X$ without any information about the bigger space $V$, in particular we do not assume that we can sum over $V$. This restriction is reasonable in many cases: if $X$ is a set of several graphs or images and $V$ is a set of all graphs or all images, then it is infeasible to sum over $V$.
>
> We are not aware of submodular functions that can be applied to our setup while being not equivalent to one of the measures in Table 1. If there is a particular function that you believe should be added to our analysis — we are happy to extend our work.
>
> > In most of the practical situations one is interested in coming up with a subset of the data starting from no datapoints.
>
> Thank you for the comment! Although our properties assume that the number of elements in a dataset is fixed, they can still be applied within iterative strategies when one adds elements successively to optimize diversity. See, e.g., the greedy algorithm in Velikonivtsev et al. (2024). Other algorithms may operate with a dataset of fixed size: we may start with a random set of objects and then use, e.g., a genetic approach (or other optimization approaches) targeting at optimizing diversity.
>
> > do these corner cases (particularly for the determinant-based examples) occur regularly?
>
> > While the work says that it is quite difficult to satisfy all three axioms, it would've been nice to discuss an existing metric that doesn't succumb to corner cases in practical situations.
>
> Please, see the first two comments in our reply to **Reviewer zmT9**. Here we provide an additional experiment and also discuss when existing measures can be used despite not having all the properties.
>
> We hope that our response addresses the raised issues.

---

> > ### Comment · Reviewer_yDDf · 2025-04-03
> >
> > # Edit for the Authors after their comment
> >
> > The latest (and biggest class) DSPNs (Bhatt, Das, and Bilmes'24) should be a better citation/discussion (this, by the way, doesn't mean you shouldn't cite the other mentioned papers, they're equally important). This class also includes how it can represent a facility location function (in the Bilmes and Bai'17).
> >
> > I'd argue that if this can't be put in the "distance" based diversity framework proposed in the paper, then it means it deserves to be discussed as a limitation of this framework. Moreover, it should also be addressed as a potential method that can be a way around the NP-Hard measures. One may be able to find a modular function (or a set of modular functions) and a concave function (or a set of concave functions) that can be made to fit well for the proposed axioms (continuity can be easily satisfied in my opinion here, however).
> >
> >
> > ## Earlier Rebuttal Response
> >
> > I thank the authors for adding the formal definitions. While one can always state things informally, formal definitions are very important.
> >
> > ## Follow-up Questions
> >
> > - I feel that in some cases diversity is very task-dependent. For instance, consider points on a hypersphere; should the diversity increase if some factor changes the radius of the hypersphere? I don't think that should always be the case (say if classification only depends on radial angle)
> >
> > -  From your reply to zmT9 --  "We see that optimizing Average (portrait_genetic_optimizing_avg.pdf) leads to more similar graph structures that tend to be either too dense or too sparse, which agrees with our observations about corner cases for Average in Section 2" -- can you please point me to how exactly this corner case is present in the average one, or in other words, which graph is the corner case that energy metric is not succumbing to? In general, I feel one can draw the same graphs in many different ways that might look very different (illusion of the eyes).
> >
> > - The mentioned greedy procedure to find a set with a high diversity value in general may not have any theoretical guarantee. Submodular functions, on the other hand, do admit a guarantee on maximization under well-behaved constraints (matroid rank, say)
> >
> >
> > ## On Submodular Functions
> > - Submodular Functions always satisfy the diminishing returns property, by definition.
> > - There exists a fairly large class of submodular functions, instantiated using features that do not need a ground set of items to be instantiated. For example, Deep Submodular Functions (Dolhansky and Bilmes'16, Bilmes and Bai'17) and its superclass Deep Submodular Peripteral Networks (Bhatt, Das and Bilmes'24), which is conjectured to represent all monotone non-decreasing normalized submodular functions.
> > - A discussion on all of the above would be good to have.
> >
> > ## References
> >
> > - Dolhansky and Bilmes'16: Deep Submodular Functions: Definitions and Learning (NeurIPS'16)
> > - Bilmes and Bai'17: Deep Submodular Functions (https://arxiv.org/abs/1701.08939)
> > - Bhatt, Das and Bilmes'24: Deep Submodular Peripteral Networks (NeurIPS'24)

---

> > > ### Author Response · Authors · 2025-04-06
> > >
> > > Thank you for your involvement in the discussion! We reply to the additional questions below.
> > >
> > > **Q: I feel that in some cases diversity is very task-dependent. For instance, consider points on a hypersphere; should the diversity increase if some factor changes the radius of the hypersphere? I don't think that should always be the case (say if classification only depends on radial angle)**
> > >
> > > Indeed, the notion of diversity can be task-dependent. In the example with a hypersphere, it is natural to choose the distance function accordingly. For instance, if classification only depends on the radial angle, one can choose the angular distance. Then, diversity would not change when we change the radius of the hypersphere. On the other hand, if we choose the Euclidean distance, then varying the hypersphere radius would change the diversity.
> > >
> > > The choice of a proper distance measure is very important. However, in this study, we assume that a distance measure suitable for a particular problem is already chosen. This allows us to keep our study task-agnostic and focus on properties that are desirable for general diversity measures.
> > >
> > > **Q: From your reply to zmT9 … can you please point me to how exactly this corner case is present in the average one, or in other words, which graph is the corner case that energy metric is not succumbing to?**
> > >
> > > We expected from our intuition and the synthetic example that Average may tend to create duplicates, especially of some “extreme” elements. In the example with generated graphs, we see that there are 9 complete graphs (all node degrees equal 15). Note that the order of graphs corresponds to decreasing density, so complete graphs are the first 9 graphs on the figure. Also, there are 7 isomorphic sparse star-shaped graphs (one central node is connected to all other nodes). Overall, for Average, among 100 generated graphs there are only 25 non-isomorphic ones. In contrast, all graphs produced by Energy are non-isomorphic, among which there is one complete graph and one empty graph.
> > >
> > > **Q: The mentioned greedy procedure to find a set with a high diversity value in general may not have any theoretical guarantee. Submodular functions, on the other hand, do admit a guarantee on maximization under well-behaved constraints (matroid rank, say).**
> > >
> > > We agree that when we limit the desirable properties to a fixed number of elements, there are no guarantees for greedy methods. Thus, other procedures can be preferred, like genetic approaches or local optimization methods. We think that it would be great to extend the list of axioms to varying dataset sizes. This can be done by adding more axioms to the current list. However, we noticed that even for the fixed size of the set, it is already extremely challenging to satisfy the desirable properties, thus we leave extending the list of axioms to future studies.
> > >
> > > **Q: There exists a fairly large class of submodular functions, instantiated using features that do not need a ground set of items to be instantiated. For example, Deep Submodular Functions (Dolhansky and Bilmes'16, Bilmes and Bai'17) and its superclass Deep Submodular Peripteral Networks (Bhatt, Das and Bilmes'24), which is conjectured to represent all monotone non-decreasing normalized submodular functions**
> > >
> > > Thank you for the references! We plan to include a discussion of submodular functions and their relation to our work in the revised version of the paper.
> > >
> > > Let us briefly discuss the referenced works. For this, let us cite Dolhansky et al. (2016):
> > >
> > > > Feature-based functions take the form $f(X) = \sum_{u \in U} w_u \phi_u\left(m_u(X)\right)$,
> > > where $\phi_u$ is a non-decreasing, non-negative, univariate, normalized concave function, $m_u(X)$ is a feature-specific non-negative modular function, and $w_u$ is a non-negative feature weight. The result is the class of feature-based submodular functions (instances of SCMs).
> > >
> > > (here $m_u: V \to \mathbb{R}$ is a non-negative modular function and $m_u(X):= \sum_{x \in X} m_u(x)$)
> > >
> > > > Another advantage of such functions is that they do not require the construction of a pairwise graph and therefore do not have quadratic cost as would, say a facility location function ... or any function based on pair-wise distances, all of which have cost $\mathcal{O}(n^2)$ to evaluate.
> > >
> > > As far as we understand, feature-based submodular functions cannot be directly applied if we want to express diversity of a set as a function of pairwise distances. Thus, we cannot analyze them in our framework. The more complex deep submodular functions are constructed in layered manner similar to neural networks, when zero layer consist of several feature-based submodular functions. Thus, deep submodular functions also do not operate with pairwise distances and cannot be analyzed within our framework.
> > >
> > > We would be happy to engage in further discussions to properly address the suggestions. Unfortunately, we are not able to post more comments, we can only edit this one.

---

### Decision · Program_Chairs · 2025-05-01

**Decision:**

Accept (poster)

**Comment:**

The authors' contribution of axioms for diversity is insightful and opens the door for future work. There was some spirited back-and-forth regarding submodular functions, and we would like the authors to acknowledge that their framework does not capture at least one class of submodular functions that are commonly used in the context of the problem they study.